# Learning from Protein Structure with Geometric Vector Perceptrons

**Bowen Jing**[*]**, Stephan Eismann**[*]**, Patricia Suriana, Raphael J.L. Townshend, Ron O. Dror**
Stanford University
{bjing, seismann, psuriana, raphael, rondror}@cs.stanford.edu

## Abstract

Learning on 3D structures of large biomolecules is emerging as a distinct area in machine learning, but there has yet to emerge a unifying network architecture that simultaneously leverages the geometric and relational aspects of the problem domain. To address this gap, we introduce *geometric vector perceptrons*, which extend standard dense layers to operate on collections of Euclidean vectors. Graph neural networks equipped with such layers are able to perform both geometric and relational reasoning on efficient representations of macromolecules. We demonstrate our approach on two important problems in learning from protein structure: model quality assessment and computational protein design. Our approach improves over existing classes of architectures on both problems, including state-of-the-art convolutional neural networks and graph neural networks. We release our code at https://github.com/drorlab/gvp.

## 1 Introduction

Many efforts in structural biology aim to predict, or derive insights from, the structure of a macro-molecule (such as a protein, RNA, or DNA), represented as a set of positions associated with atoms or groups of atoms in 3D Euclidean space. These problems can often be framed as functions mapping the input domain of structures to some property of interest—for example, predicting the quality of a structural model or determining whether two molecules will bind in a particular geometry. Thanks to their importance and difficulty, such problems, which we broadly refer to as *learning from structure*, have recently developed into an exciting and promising application area for deep learning (Graves et al., 2020; Ingraham et al., 2019; Pereira et al., 2016; Townshend et al., 2019; Won et al., 2019).

Successful applications of deep learning are often driven by techniques that leverage the problem structure of the domain—for example, convolutions in computer vision (Cohen & Shashua, 2017) and attention in natural language processing (Vaswani et al., 2017). What are the relevant considerations in the domain of learning from structure? Using proteins as the most common example, we have on the one hand the arrangement and orientation of the amino acid residues in space, which govern the dynamics and function of the molecule (Berg et al., 2002). On the other hand, proteins also possess relational structure in terms of their amino-acid sequence and the residue-residue interactions that mediate the aforementioned protein properties (Hammes-Schiffer & Benkovic, 2006). We refer to these as the *geometric* and *relational* aspects of the problem domain, respectively.

Recent state-of-the-art methods for learning from structure leverage one of these two aspects. Commonly, such methods employ either graph neural networks (GNNs), which are expressive in terms of relational reasoning (Battaglia et al., 2018), or convolutional neural networks (CNNs), which operate directly on the geometry of the structure. Here, we present a unifying architecture that bridges these two families of methods to leverage *both* aspects of the problem domain.

We do so by introducing *geometric vector perceptrons* (GVPs), a drop-in replacement for standard multi-layer perceptrons (MLPs) in aggregation and feed-forward layers of GNNs. GVPs operate directly on both scalar and *geometric* features—features that transform as a vector under a rotation of spatial coordinates. GVPs therefore allow for the embedding of geometric information at nodes and

---

[*]Equal contribution

edges without reducing such information to scalars that may not fully capture complex geometry. We postulate that our approach makes it easier for a GNN to learn functions whose significant features are both geometric and relational.

Our method (GVP-GNN) can be applied to any problem where the input domain is a structure of a single macromolecule or of molecules bound to one another. In this work, we specifically demonstrate our approach on two problems connected to protein structure: computational protein design and model quality assessment. Computational protein design (CPD) is the conceptual inverse of protein structure prediction, aiming to infer an amino acid sequence that will fold into a given structure. Model quality assessment (MQA) aims to select the best structural model of a protein from a large pool of candidate structures and is an important step in structure prediction (Cheng et al., 2019). Our method outperforms existing methods on both tasks.

## 2 RELATED WORK

ML methods for learning from protein structure largely fall into one of three types, operating on *sequential*, *voxelized*, or *graph-structured* representations of proteins. We briefly discuss each type and introduce state-of-the-art examples for MQA and CPD to set the stage for our experiments later.

**Sequential representations** In traditional models of learning from protein structure, each amino acid is represented as a feature vector using hand-crafted representations of the 3D structural environment. These representations include residue contacts (Olechnovič & Venclovas, 2017), orientations or positions collectively projected to local coordinates (Karasikov et al., 2019), physics-inspired energy terms (O'Connell et al., 2018; Uziela et al., 2017), or context-free grammars of protein topology (Greener et al., 2018). The structure is then viewed as a sequence or collection of such features which can be fed into a 1D convolutional network, RNN, or dense feedforward network. Although these methods only indirectly represent the full 3D structure of the protein, a number of them, such as ProQ4 (Hurtado et al., 2018), VoroMQA (Olechnovič & Venclovas, 2017), and SBROD (Karasikov et al., 2019), are competitive in assessments of MQA.

**Voxelized representations** In lieu of hand-crafted representations of structure, 3D convolutional neural networks (CNNs) can operate directly on the positions of atoms in space, encoded as occupancy maps in a voxelized 3D volume. The hierarchical convolutions of such networks are easily compatible with the detection of structural motifs, binding pockets, and the specific shapes of other important structural features, leveraging the *geometric* aspect of the domain. A number of CPD methods (Anand et al., 2020; Zhang et al., 2019; Shroff et al., 2019) and the MQA methods 3DCNN (Derevyanko et al., 2018) and Ornate (Pagès et al., 2019) exemplify the power of this approach.

**Graph-structured representations** A protein structure can also be represented as a proximity graph over amino acid nodes, reducing the challenge of representing a collective structural neighborhood in a single feature vector to that of representing individual edges. Graph neural networks (GNNs) can then perform complex *relational* reasoning over structures (Battaglia et al., 2018)—for example, identifying key relationships among amino acids, or flexible structural motifs described as a connectivity pattern rather than a rigid shape. Recent state-of-the-art GNNs include Structured Transformer (Ingraham et al., 2019) on CPD, ProteinSolver (Strokach et al., 2020) on CPD and mutation stability prediction, and GraphQA (Baldassarre et al., 2020) on MQA. These methods vary in their representation of geometry: while some, such as ProteinSolver and GraphQA, represent edges as a function of their length, others, such as Structured Transformer, indirectly encode the 3D geometry of the proximity graph in terms of relative orientations and other scalar features.

## 3 METHODS

Our architecture seeks to combine the strengths of CNN and GNN methods in learning from biomolecular structure by improving the latter's ability to reason *geometrically*. The GNNs described in the previous section encode the 3D geometry of the protein by encoding *vector* features (such as node orientations and edge directions) in terms of rotation-invariant *scalars*, often by defining a local coordinate system at each node. We instead propose that these features be *directly*

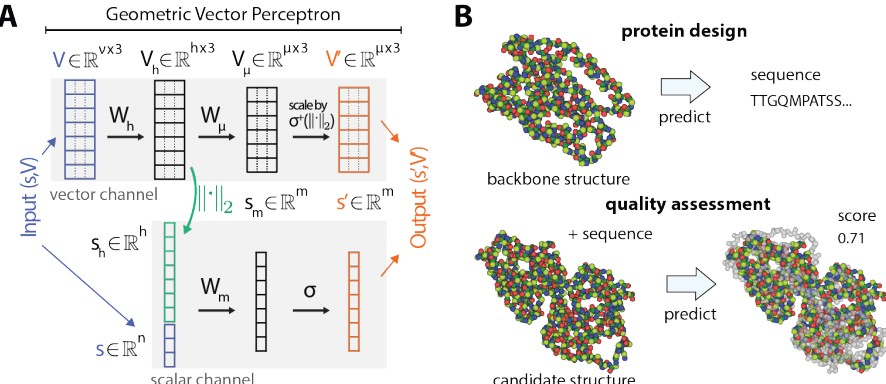

Figure 1: **(A)** Schematic of the geometric vector perceptron illustrating Algorithm 1. Given a tuple of scalar and vector input features $(\mathbf{s}, \mathbf{V})$, the perceptron computes an updated tuple $(\mathbf{s}', \mathbf{V}')$. $\mathbf{s}'$ is a function of both $\mathbf{s}$ and $\mathbf{V}$. **(B)** Illustration of the structure-based prediction tasks. In computational protein design (top), the goal is to predict an amino acid sequence that would fold into a given protein backbone structure. Individual atoms are represented as colored spheres. In model quality assessment (bottom), the goal is to predict the quality score of a candidate structure, which measures the similarity of the candidate with respect to the experimentally determined structure (in gray).

*represented as geometric vectors*—features in $\mathbb{R}^3$ which transform appropriately under a change of spatial coordinates—at all steps of graph propagation.

This conceptual shift has two important ramifications. First, the input representation is more efficient: instead of encoding the orientation of a node by its relative orientation with all of its neighbors, we only have to represent one absolute orientation per node. Second, it standardizes a global coordinate system across the entire structure, which allows geometric features to be directly propagated without transforming between local coordinates. For example, representations of arbitrary positions in space—including points that are not themselves nodes—can be easily propagated across the graph by Euclidean vector addition. We postulate this allows the GNN to more easily access global geometric properties of the structure. The key challenge with this representation, however, is to perform graph propagation in a way that simultaneously preserves the full expressive power of the original GNN while maintaining the rotation invariance provided by the scalar representations. We do so by introducing a new module, the *geometric vector perceptron*, to replace dense layers in a GNN.

## 3.1 GEOMETRIC VECTOR PERCEPTRONS

The geometric vector perceptron is a simple module for learning vector-valued and scalar-valued functions over geometric vectors and scalars. That is, given a tuple $(\mathbf{s}, \mathbf{V})$ of scalar features $\mathbf{s} \in \mathbb{R}^n$ and vector features $\mathbf{V} \in \mathbb{R}^{\nu \times 3}$, we compute new features $(\mathbf{s}', \mathbf{V}') \in \mathbb{R}^m \times \mathbb{R}^{\mu \times 3}$. The computation is illustrated in Figure 1A and formally described in Algorithm 1.

At its core, the GVP consists of two separate linear transformations $\mathbf{W}_m, \mathbf{W}_h$ for the scalar and vector features, followed by nonlinearities $\sigma, \sigma^+$. However, before the scalar features are transformed, we concatenate the $L_2$ norm of the transformed vector features $\mathbf{V}_h$; this allows us to extract rotation-invariant information from the input vectors $\mathbf{V}$. An additional linear transformation $\mathbf{W}_\mu$ is inserted just before the vector nonlinearity to control the output dimensionality independently of the number of norms extracted.

The GVP is conceptually simple, yet provably possesses the desired properties of invariance/equivariance and expressiveness. First, the vector and scalar outputs of the GVP are equivariant and invariant, respectively, with respect to an arbitrary composition $R$ of rotations and reflections in 3D Euclidean space — *i.e.*, if GVP$(\mathbf{s}, \mathbf{V}) = (\mathbf{s}', \mathbf{V}')$ then

$$\text{GVP}(\mathbf{s}, R(\mathbf{V})) = (\mathbf{s}', R(\mathbf{V}'))  \tag{1}$$

---

**Algorithm 1** Geometric vector perceptron

---

**Input**: Scalar and vector features $(\mathbf{s}, \mathbf{V}) \in \mathbb{R}^n \times \mathbb{R}^{\nu \times 3}$ .
**Output**: Scalar and vector features $(\mathbf{s}', \mathbf{V}') \in \mathbb{R}^m \times \mathbb{R}^{\mu \times 3}$ .
$h \leftarrow \max(\nu, \mu)$
**GVP**:
   $\mathbf{V}_h \leftarrow \mathbf{W}_h \mathbf{V} \quad \in \mathbb{R}^{h \times 3}$
   $\mathbf{V}_\mu \leftarrow \mathbf{W}_\mu \mathbf{V}_h \quad \in \mathbb{R}^{\mu \times 3}$
   $\mathbf{s}_h \leftarrow \|\mathbf{V}_h\|_2 \text{ (row-wise)} \quad \in \mathbb{R}^h$
   $\mathbf{v}_\mu \leftarrow \|\mathbf{V}_\mu\|_2 \text{ (row-wise)} \quad \in \mathbb{R}^\mu$
   $\mathbf{s}_{h+n} \leftarrow \text{concat}(\mathbf{s}_h, \mathbf{s}) \quad \in \mathbb{R}^{h+n}$
   $\mathbf{s}_m \leftarrow \mathbf{W}_m \mathbf{s}_{h+n} + \mathbf{b} \quad \in \mathbb{R}^m$
   $\mathbf{s}' \leftarrow \sigma(\mathbf{s}_m) \quad \in \mathbb{R}^m$
   $\mathbf{V}' \leftarrow \sigma^+(\mathbf{v}_\mu) \odot \mathbf{V}_\mu \text{ (row-wise multiplication)} \quad \in \mathbb{R}^{\mu \times 3}$
**return** $(\mathbf{s}', \mathbf{V}')$

---

This is due to the fact that the only operations on vector-valued inputs are scalar multiplication, linear combination, and the $L_2$ norm.[1] We include a formal proof in Appendix A.

In addition, the GVP architecture can approximate any continuous rotation- and reflection-invariant scalar-valued function of $\mathbf{V}$. More precisely, let $G_s$ be a GVP defined with $n, \mu = 0$—that is, one which transforms vector features to scalar features. Then for any function $f : \mathbb{R}^{\nu \times 3} \to \mathbb{R}$ invariant with respect to rotations and reflections in 3D, there exists a functional form $G_s$ able to $\epsilon$-approximate $f$, given mild assumptions.

**Theorem.** *Let $R$ describe an arbitrary rotation and/or reflection in $\mathbb{R}^3$. For $\nu \geq 3$ let $\Omega^\nu \subset \mathbb{R}^{\nu \times 3}$ be the set of all $\mathbf{V} = [\mathbf{v}_1, \quad \ldots, \quad \mathbf{v}_\nu]^T \in \mathbb{R}^{\nu \times 3}$ such that $\mathbf{v}_1, \mathbf{v}_2, \mathbf{v}_3$ are linearly independent and $0 < \|\mathbf{v}_i\|_2 \leq b$ for all $i$ and some finite $b > 0$. Then for any continuous $F : \Omega^\nu \to \mathbb{R}$ such that $F(R(\mathbf{V})) = F(\mathbf{V})$ and for any $\epsilon > 0$, there exists a form $f(\mathbf{V}) = \mathbf{w}^T G_s(\mathbf{V})$ such that $|F(\mathbf{V}) - f(\mathbf{V})| < \epsilon$ for all $\mathbf{V} \in \Omega^\nu$.*

We include a formal proof in Appendix A. As a corollary, a GVP with nonzero $n, \mu$ is also able to approximate similarly-defined functions over the full input domain $\mathbb{R}^n \times \mathbb{R}^{\nu \times 3}$.

In addition to the GVP layer itself, we use a version of dropout that drops entire vector channels at random (as opposed to coordinates within vector channels). We also introduce layer normalization for the vector features as

$$\mathbf{V} \leftarrow \mathbf{V} / \sqrt{\frac{1}{\nu} \|\mathbf{V}\|_2^2} \quad \in \mathbb{R}^{\nu \times 3} \tag{2}$$

That is, we scale the row vectors of $\mathbf{V}$ such that their root-mean-square norm is one. This vector layer norm has no trainable parameters, but we continue to use normal layer normalization on scalar channels with trainable parameters $\gamma, \beta$.

We study our hypothesis that GVPs augment the geometric reasoning ability of GNNs on a synthetic dataset (Appendix B). The synthetic dataset allows us to control the function underlying the ground-truth label in order to explicitly separate geometric and relational aspects in different tasks. The GVP-augmented GNN (or GVP-GNN) matches a CNN on a geometric task and a standard GNN on a relational task. However, when we combine the two tasks in one objective, the GVP-GNN does significantly better than either a GNN or a CNN.

### 3.2 REPRESENTATIONS OF PROTEINS

The main empirical validation of our architecture is its performance on two real-world tasks: computational protein design (CPD) and model quality assessment (MQA). These tasks, as illustrated in Figure 1B and described in detail in Section 4, are complementary in that one (CPD) predicts a property for each amino acid while the other (MQA) predicts a global property.

---

[1]The nonlinearity $\sigma^+$ is a *scaling* by $\sigma^+$ applied to the $L_2$ norm.

We represent a protein structure input as a proximity graph with a minimal number of scalar and vector features to specify the 3D structure of the molecule. A protein structure is a sequence of amino acids, where each amino acid consists of four *backbone* atoms[2] and a set of *sidechain* atoms located in 3D Euclidean space. We represent only the backbone because the sidechains are unknown in CPD, and our MQA benchmark corresponds to the assessment of backbone structure only.

Let $X_i$ be the position of atom X in the $i$th amino acid (e.g. $N_i$ is the position of the nitrogen atom in the $i$th amino acid). We represent backbone structure as a graph $\mathcal{G} = (\mathcal{V}, \mathcal{E})$ where each node $\mathfrak{v}_i \in \mathcal{V}$ corresponds to an amino acid and has embedding $\mathbf{h}_{\mathfrak{v}}^{(i)}$ with the following features:

- Scalar features $\{\sin, \cos\} \circ \{\phi, \psi, \omega\}$, where $\phi, \psi, \omega$ are the dihedral angles computed from $C_{i-1}, N_i, C\alpha_i, C_i$, and $N_{i+1}$.
- The *forward* and *reverse* unit vectors in the directions of $C\alpha_{i+1} - C\alpha_i$ and $C\alpha_{i-1} - C\alpha_i$, respectively.
- The unit vector in the imputed direction of $C\beta_i - C\alpha_i$.[3] This is computed by assuming tetrahedral geometry and normalizing

$$\sqrt{\frac{1}{3}}(\mathbf{n} \times \mathbf{c})/||\mathbf{n} \times \mathbf{c}||_2 - \sqrt{\frac{2}{3}}(\mathbf{n} + \mathbf{c})/||\mathbf{n} + \mathbf{c}||_2$$

  where $\mathbf{n} = N_i - C\alpha_i$ and $\mathbf{c} = C_i - C\alpha_i$. This vector, along with the forward and reverse unit vectors, unambiguously define the orientation of each amino acid residue.
- A one-hot representation of amino acid identity, when available.

The set of edges is $\mathcal{E} = \{\mathbf{e}_{j \to i}\}_{i \neq j}$ for all $i, j$ where $\mathfrak{v}_j$ is among the $k = 30$ nearest neighbors of $\mathfrak{v}_i$ as measured by the distance between their $C\alpha$ atoms. Each edge has an embedding $\mathbf{h}_e^{(j \to i)}$ with the following features:

- The unit vector in the direction of $C\alpha_j - C\alpha_i$.
- The encoding of the distance $||C\alpha_j - C\alpha_i||_2$ in terms of Gaussian radial basis functions.[4]
- A sinusoidal encoding of $j - i$ as described in Vaswani et al. (2017), representing distance along the backbone.

In our notation, each feature vector $\mathbf{h}$ is a concatenation of scalar and vector features as described above. Collectively, these features are sufficient for a complete description of the protein backbone.

### 3.3 NETWORK ARCHITECTURE

Our architecture (GVP-GNN) leverages message passing (Gilmer et al., 2017) in which messages from neighboring nodes and edges are used to update node embeddings at each graph propagation step. More explicitly, the architecture takes as input the protein graph defined above and performs graph propagation steps according to:

$$\mathbf{h}_m^{(j \to i)} := g\left(\text{concat}\left(\mathbf{h}_{\mathfrak{v}}^{(j)}, \mathbf{h}_e^{(j \to i)}\right)\right) \tag{3}$$

$$\mathbf{h}_{\mathfrak{v}}^{(i)} \leftarrow \text{LayerNorm}\left(\mathbf{h}_{\mathfrak{v}}^{(i)} + \frac{1}{k'}\text{Dropout}\left(\sum_{j:\mathbf{e}_{j \to i} \in \mathcal{E}} \mathbf{h}_m^{(j \to i)}\right)\right) \tag{4}$$

Here, $g$ is a sequence of three GVPs, $\mathbf{h}_{\mathfrak{v}}^{(i)}$ and $\mathbf{h}_e^{(j \to i)}$ are the embeddings of the node $i$ and edge $(j \to i)$ as above, and $\mathbf{h}_m^{(j \to i)}$ represents the message passed from node $j$ to node $i$. $k'$ is the number of incoming messages, which is equal to $k$ unless the protein contains fewer than $k$ amino acid residues. Between graph propagation steps, we also use a feed-forward point-wise layer to update the node embeddings at all nodes $i$:

$$\mathbf{h}_{\mathfrak{v}}^{(i)} \leftarrow \text{LayerNorm}\left(\mathbf{h}_{\mathfrak{v}}^{(i)} + \text{Dropout}\left(g\left(\mathbf{h}_{\mathfrak{v}}^{(i)}\right)\right)\right) \tag{5}$$

---

[2] $C\alpha$, C, N, and O. The alpha carbon $C\alpha$ is the central carbon atom in each amino acid residue.
[3] $C\beta$ is the second carbon from the carboxyl carbon C.
[4] We use 16 Gaussian radial basis functions with centers evenly spaced between 0 and 20 angstroms.

where $g$ is a sequence of two GVPs. These graph propagation and feed-forward steps update the vector features at each node in addition to its scalar features.

In computational protein design, the network learns a generative model over the space of protein sequences conditioned on the given backbone structure. Following Ingraham et al. (2019), we frame this as an *autoregressive* task and use a masked encoder-decoder architecture to capture the joint distribution over all positions: for each $i$, the network models the distribution at $i$ based on the complete structure graph, as well as the sequence information at positions $j < i$. The encoder first performs three graph propagation steps on the structural information only. Then, sequence information is added to the graph, and the decoder performs three further graph propagation steps where incoming messages $\mathbf{h}_m^{(j \to i)}$ for $j \geq i$ are computed only with the encoder embeddings. Finally, we use one last GVP with 20-way scalar softmax output to predict the probability of the amino acids.

In model quality assessment, we use three graph propagation steps and perform regression against the true quality score of a candidate structure, a global scalar property. To obtain a single global representation, we apply a node-wise GVP to reduce all node embeddings to scalars. We then average the representations across all nodes and apply a final dense feed-forward network to output the network's prediction.

Further details regarding training and hyperparameters can be found in Appendix D.

## 4 EVALUATION METRICS AND DATASETS

**Protein design** Computational protein design (CPD) is the conceptual inverse of protein structure prediction, aiming to infer an amino acid sequence that will fold into a given structure. CPD is difficult to directly benchmark, as some structures may correspond to a large space of sequences and others may correspond to none at all. Therefore, the proxy metric of *native sequence recovery*—inferring native sequences given their experimentally determined structures—is often used (Li et al., 2014; O'Connell et al., 2018; Wang et al., 2018). Drawing an analogy between sequence design and language modelling, Ingraham et al. (2019) also evaluate the model *perplexity* on held-out native sequences. Both metrics rest on the implicit assumption that native sequences are optimized for their structures (Kuhlman & Baker, 2000) and should be assigned high probabilities.

To best approximate real-world applications that may require design of novel structures, the held-out evaluation set should bear minimal similarity to the training structures. We use the CATH 4.2 dataset curated by Ingraham et al. (2019) in which all available structures with 40% nonredudancy are partitioned by their CATH (class, architecture, topology/fold, homologous superfamily) classification. The training, validation, and test splits consist of 18204, 608, and 1120 structures, respectively.

We also report results on TS50, an older test set of 50 native structures first introduced by Li et al. (2014). The smaller size of this benchmark also allows a comparison to the computationally expensive physics-based calculations of the `fixbb` protocol in Rosetta, a software suite well-established in the structural biology community (Das & Baker, 2008). No canonical training and validation sets exist for TS50. To evaluate on TS50, we filter the CATH 4.2 training and validation sets for sequences with less than 30% similarity (as computed by `PSIBLAST`) to any sequence in TS50.

**Model quality assessment** Model quality assessment (MQA) aims to select the best structural model of a protein from a large pool of candidate structures.[5] The performance of different MQA methods is evaluated every two years in the community-wide Critical Assessment of Structure Prediction (CASP) (Cheng et al., 2019). For a number of recently solved but unreleased structures, called *targets*, structure generation programs produce a large number of candidate structures. MQA methods are evaluated by how well they predict the GDT-TS score of a candidate structure compared to the experimentally solved structure for that target. GDT-TS is a scalar measure of how similar two protein backbones are after global alignment (Zemla et al., 2001).

In addition to accurately predicting the absolute quality of a candidate structure, a good MQA method should also be able to accurately assess the relative model qualities among a pool of candidates for a given target so that the best ones can be selected, perhaps for further refinement. There-

---

[5]We refer to models of protein structure as "structural models" or "candidate structures" to avoid confusion with the term "model" as used in the ML community.

Table 1: GVP-GNN outperforms Structured Transformer and sets a new state-of-the art on the CATH 4.2 protein design test set (and its short and single-chain subsets) in terms of per-residue perplexity (lower is better) and recovery (higher is better). Recovery is reported as the median (over all structures) of the average % of residues correctly recovered in 100 sampled sequences.

| | | Perplexity | | | Recovery % | | |
|---|---|---|---|---|---|---|---|
| Method | Type | Short | Single-chain | All | Short | Single-chain | All |
| GVP-GNN | GNN | **7.10** | **7.44** | **5.29** | **32.1** | **32.0** | **40.2** |
| Structured GNN | GNN | 8.31 | 8.88 | 6.55 | 28.4 | 28.1 | 37.3 |
| Structured Transformer | GNN | 8.54 | 9.03 | 6.85 | 28.3 | 27.6 | 36.4 |

fore, MQA methods are commonly evaluated on two metrics: a *global* correlation between the predicted and ground truth scores, pooled across all targets, and the average *per-target* correlation among only the candidate structures for a specific target (Cao & Cheng, 2016; Derevyanko et al., 2018; Pagès et al., 2019). We follow this convention in our experiments.

We train and validate on 79200 candidate structures for 528 targets submitted to CASP 5-10. We then test GVP-GNN on two MQA datasets. First, we score 20880 stage 1 and stage 2 candidate structures from CASP 11 (84 targets) and 12 (40 targets). This benchmark was first established by Karasikov et al. (2019) and has been used by many recently published methods. Second, to compare with a larger number of methods on more recent structural data, we also score 1472 stage 2 candidate structures from CASP 13 (20 targets). We add the CASP 11-12 structures to our training set to evaluate on CASP 13. Further details on the MQA datasets can be found in Appendix C.

## 5 EXPERIMENTS

**Protein design**  GVP-GNN achieves state-of-the-art performance on CATH 4.2, representing a substantial improvement both in terms of perplexity and sequence recovery over Structured Transformer (Ingraham et al., 2019), a GNN method which was trained using the same training and validation sets (Table 1). Following Ingraham et al. (2019), we report evaluation on short (100 or fewer amino acid residues) and single-chain subsets of the CATH 4.2 test set, containing 94 and 103 proteins, respectively, in addition to the full test set. Although Structured Transformer leverages an attention mechanism on top of a graph-structured representation of proteins, the authors note in ablation studies that removing attention appeared to increase performance. We therefore retrain and compare against a version of Structured Transformer with the attention layers replaced with standard graph propagation operations (Structured GNN). Our method also improves upon this model.

On the smaller test set TS50, we achieve 44.9% recovery compared to Rosetta's 30% and outperform methods based on each of the three classes of structural representations. Overall, we place 2nd out of 9 methods in terms of recovery (see Appendix E). However, the results for this test set should be taken with a grain of salt, given that the different methods did not use canonical training datasets.

**Model quality assessment**  We compare GVP-GNN against other single-structure, structure-only methods on the CASP 11-12 test set in Table 2.[6]  These include the CNN methods 3DCNN (Derevyanko et al., 2018) and Ornate (Pagès et al., 2019), the GNN method GraphQA (Baldassarre et al., 2020), and three methods that use sequential representations—VoroMQA (Olechnovič & Venclovas, 2017), SBROD (Karasikov et al., 2019), and ProQ3D (Uziela et al., 2017). All of these methods learn solely from protein structure,[7] with the exception of ProQ3D, which in addition uses sequence profiles based on alignments. We include ProQ3D because it is an improved version of the best single-model method in CASP 11 and CASP 12 (Uziela et al., 2017). GVP-GNN outperforms all other structural methods in both global and per-target correlation, and even performs better than ProQ3D on all but one benchmark. We also train and evaluate DimeNet, a recent 3D-aware GNN architecture which achieves state-of-the-art on many small-molecule tasks (Klicpera et al., 2019), on CASP 11-12. DimeNet does not outperform any of the models in Table 2 (see Appendix E).

---

[6]We obtained the data for the comparisons from Baldassarre et al. (2019).

[7]There are two versions of GraphQA; we compare against the one using only structure information.

Table 2: GVP-GNN improves over other single-structure, structure-only methods on CASP 11 and 12 in terms of global (Glob) and mean per-target (Per) Pearson correlation coefficients (higher is better). Each method is classified as one of the three types discussed in Section 2. ProQ3D is set aside as the only method shown which additionally uses sequence-based profiles. For each metric, the top performing structure-only method is in bold, as is the top method overall (if different).

| | | CASP 11 | | | | CASP 12 | | | |
| | | Stage 1 | | Stage 2 | | Stage 1 | | Stage 2 | |
| Method | Type | Glob | Per | Glob | Per | Glob | Per | Glob | Per |
|---|---|---|---|---|---|---|---|---|---|
| GVP-GNN | GNN | **0.84** | **0.66** | **0.87** | **0.45** | **0.79** | **0.73** | **0.82** | **0.62** |
| 3DCNN | CNN | 0.59 | 0.52 | 0.64 | 0.40 | 0.49 | 0.44 | 0.61 | 0.51 |
| Ornate | CNN | 0.64 | 0.47 | 0.63 | 0.39 | 0.55 | 0.57 | 0.67 | 0.49 |
| GraphQA | GNN | 0.83 | 0.63 | 0.82 | 0.38 | 0.72 | 0.68 | 0.81 | 0.61 |
| VoroMQA | Seq | 0.69 | 0.62 | 0.65 | 0.42 | 0.46 | 0.61 | 0.61 | 0.56 |
| SBROD | Seq | 0.58 | 0.65 | 0.55 | 0.43 | 0.37 | 0.64 | 0.47 | 0.61 |
| ProQ3D | Seq | 0.80 | **0.69** | 0.77 | 0.44 | 0.67 | 0.71 | 0.81 | 0.60 |

Table 3: GVP-GNN improves over single-structure methods participating in CASP 13 on the 20 evaluated targets. The seven top methods highlighted by the CASP organizers are shown. GVP-GNN is the top structure-only method and the top method overall in terms of global correlation.

| Method | Global | Per-target |
|---|---|---|
| GVP-GNN | **0.888** | **0.671** |
| SASHAN | 0.840 | 0.633 |
| FaeNNz | 0.810 | 0.650 |
| VoroMQA-A | 0.744 | 0.595 |
| VoroMQA-B | 0.726 | 0.586 |
| ProQ3D | 0.847 | 0.660 |
| MULTICOM-NOVEL | 0.652 | 0.551 |
| ProQ4 | 0.604 | **0.691** |

We compare GVP-GNN with all 23 single-structure MQA methods participating in CASP 13 for which complete predictions on the 20 evaluation targets are available. Seven of these methods were highlighted as best-performing by the CASP organizers in Cheng et al. (2019) and are shown along with GVP-GNN in Table 3. These include four methods learning solely from structural features and three also using sequence profiles. SASHAN learns a linear model over secondary structure and contact-based features (Cheng et al., 2019). FaeNNz[8] (Studer et al., 2020), ProQ3D (Uziela et al., 2017), and VoroMQA[9] (Olechnovič & Venclovas, 2017) learn a multi-layer perceptron or statistical potential on top of such structural features. Finally, MULTICOM-NOVEL (Hou et al., 2019) and ProQ4 (Hurtado et al., 2018) employ one-dimensional deep convolutional networks on top of sequential representations. GVP-GNN outperforms all methods in terms of global correlation and outperforms all structure-only methods in per-target correlation. See Appendix E for full results.

Finally, because our architecture updates vector features along with scalar features at each node embedding, it is possible to visualize *learned* vector features in the intermediate layers of the trained MQA network. We show and discuss the interpretability of such features in Appendix F.

**Ablation studies** The methods we have compared against include a number of GNNs (Structured Transformer/GNN, ProteinSolver, GraphQA). We train a number of ablated models for CPD and MQA to identify the aspects of the GVP which most contribute to our performance improvement over these GNNs (Table 4). Replacing the GVP with a vanilla MLP layer or propagating only scalar features both remove direct access to geometric information, forcing the model to learn scalar-

---

[8]FaeNNz is also known as QMEANDisCo
[9]VoroMQA-A and VoroMQA-B are the same except that the former relaxes the sidechains before scoring.

Table 4: Ablations of the GVP architecture decrease performance on CPD and MQA. We include Structured GNN and GraphQA as state-of-the-art GNN references for CPD and MQA, respectively. Metrics are defined the same way as in Tables 1 (CPD) and 2 (MQA).

| | CPD | | MQA | | | |
| | CATH 4.2 All | | CASP 11 Stage 2 | | CASP 12 Stage 2 | |
| Modification | Perplexity | Recovery | Global | Per-target | Global | Per-target |
|---|---|---|---|---|---|---|
| None | **5.29** | **40.2** | **0.87** | **0.45** | 0.82 | **0.62** |
| MLP layer | 7.76 | 30.6 | 0.84 | 0.36 | 0.79 | 0.59 |
| Only scalars | 7.31 | 32.4 | 0.84 | 0.38 | **0.83** | 0.59 |
| Only vectors | 11.05 | 23.2 | 0.56 | 0.16 | 0.57 | 0.39 |
| No $\mathbf{W}_\mu$ | 5.85 | 37.1 | 0.86 | 0.41 | 0.81 | 0.60 |
| Structured GNN | 6.55 | 37.3 | – | – | – | – |
| GraphQA | – | – | 0.82 | 0.38 | 0.81 | 0.61 |

valued, indirect representations of geometry. These modifications result in considerable decreases in performance, underscoring the importance of direct access to geometric information. Propagating only the vector features results in an even larger decrease as it both eliminates important scalar input features (such as torsion angles and amino acid identity) and the part of the GVP with approximation guarantees. Therefore, the dual scalar/vector design of the GVP is essential: without either, the best ablated model falls short of Structured GNN on CPD and only matches GraphQA on MQA. Finally, eliminating the second vector transformation $\mathbf{W}_\mu$ results in a slight decrease in performance. Therefore, all architectural elements contributed to our improvement over state-of-the-art.

## 6 CONCLUSION

In this work, we developed the first architecture designed specifically for learning on dual relational and geometric representations of 3D macromolecular structure. At its core, our method, GVP-GNN, augments graph neural networks with computationally simple layers that perform expressive geometric reasoning over Euclidean vector features. Our method possesses desirable theoretical properties and empirically outperforms existing architectures on learning quality scores and sequence designs, respectively, from protein structure.

The equivariance of GVP layers with respect to 3D translations and rotations also highlights a similarity to methods that leverage irreducible representations of $SO(3)$ to define equivariant convolutions on point clouds (Thomas et al., 2018; Anderson et al., 2019). These methods allow for equivariant representations of higher-order tensors, but due to their complexity and computational cost, their applications have until recently been limited to small molecules (Eismann et al., 2020). Our architecture presents an alternative, relatively lightweight approach to equivariance that is well-suited for large biomolecules and biomolecular complexes.

In further work, we hope to apply our architecture to other important structural biology problem areas, including protein complexes, RNA structure, and protein-ligand interactions.

## ACKNOWLEDGEMENTS

We acknowledge support from the U.S. Department of Energy, Office of Science, Office of Advanced Scientific Computing Research, Scientific Discovery through Advanced Computing (SciDAC) program, and Intel Corporation. SE is supported by a Stanford Bio-X Bowes fellowship. RJLT is supported by the U.S. Department of Energy, Office of Science Graduate Student Research (SCGSR) program. We thank Tri Dao, Trenton Chang, and all members of the Dror group for feedback and discussions.

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

## A  PROPERTIES OF GEOMETRIC VECTOR PERCEPTRONS

### A.1  EQUIVARIANCE AND INVARIANCE

The vector and scalar outputs of the GVP are equivariant and invariant, respectively, with respect to an arbitrary composition of rotations and reflections in 3D Euclidean space described by $R$ *i.e.*,

$$\text{GVP}((\mathbf{s}, R(\mathbf{V}))) = (\mathbf{s}', R(\mathbf{V}'))$$

*Proof.* We can write the transformation described by $R$ as multiplying $\mathbf{V}$ with a unitary matrix $\mathbf{U} \in \mathbb{R}^{3 \times 3}$ from the right. The L$_2$-norm, scalar multiplications, and nonlinearities are defined row-wise as in Algorithm 1. We consider scalar and vector outputs separately. The scalar output, as a function of the inputs, is

$$\mathbf{s}' = \sigma \left( \mathbf{W}_m \begin{bmatrix} \|\mathbf{W}_h \mathbf{V}\|_2 \\ \mathbf{s} \end{bmatrix} + \mathbf{b} \right)$$

Since $\|\mathbf{W}_h \mathbf{V} \mathbf{U}\|_2 = \|\mathbf{W}_h \mathbf{V}\|_2$, we conclude $\mathbf{s}'$ is invariant. Similarly the vector output is

$$\mathbf{V}' = \sigma^+ \left( \|\mathbf{W}_\mu \mathbf{W}_h \mathbf{V}\|_2 \right) \odot \mathbf{W}_\mu \mathbf{W}_h \mathbf{V}$$

The row-wise scaling can also be viewed as left-multiplication by a diagonal matrix $\mathbf{D}$. Since $\|\mathbf{W}_\mu \mathbf{W}_h \mathbf{V}\|_2 = \|\mathbf{W}_\mu \mathbf{W}_h \mathbf{V} \mathbf{U}\|_2$, $\mathbf{D}$ is invariant. Since

$$\mathbf{D} \mathbf{W}_\mu \mathbf{W}_h (\mathbf{V} \mathbf{U}) = (\mathbf{D} \mathbf{W}_\mu \mathbf{W}_h \mathbf{V}) \, \mathbf{U}$$

we conclude that $\mathbf{V}'$ is equivariant. □

### A.2  APPROXIMATION OF ROTATION-INVARIANT FUNCTIONS

The GVP inherits an analogue of the Universal Approximation property Cybenko (1989) of standard dense layers. If $R$ describes an arbitrary rotation or reflection in 3D Euclidean space, we show that the GVP architecture can approximate arbitrary scalar-valued functions invariant under $R$ and defined over $\Omega^\nu \subset \mathbb{R}^{\nu \times 3}$, the bounded subset of $\mathbb{R}^{\nu \times 3}$ whose elements can be canonically oriented based on three linearly independent vector entries. Without loss of generality, we assume the first three vector entries can be used.

The machinery corresponding to such approximations corresponds to a GVP $G_s$ with only vector inputs, only scalar outputs, and a sigmoidal nonlinearity $\sigma$; followed by a dense layer. This can also be viewed as the sequence of matrix multiplication with $\mathbf{W}_h$, taking the L$_2$-norm, and a dense network with one hidden layer. Such machinery can be extracted from any two consecutive GVPs (assuming a sigmoidal $\sigma$).

We restate the theorem from the main text:

**Theorem.** *Let R describe an arbitrary rotation and/or reflection in $\mathbb{R}^3$. For $\nu \geq 3$ let $\Omega^\nu \subset \mathbb{R}^{\nu \times 3}$ be the set of all $\mathbf{V} = [\boldsymbol{v}_1, \quad \ldots, \quad \boldsymbol{v}_\nu]^T \in \mathbb{R}^{\nu \times 3}$ such that $\boldsymbol{v}_1, \boldsymbol{v}_2, \boldsymbol{v}_3$ are linearly independent and $0 < \|\boldsymbol{v}_i\|_2 \leq b$ for all $i$ and some finite $b > 0$. Then for any continuous $F : \Omega^\nu \to \mathbb{R}$ such that $F(R(\mathbf{V})) = F(\mathbf{V})$ and for any $\epsilon > 0$, there exists a form $f(\mathbf{V}) = \mathbf{w}^T G_s(\mathbf{V})$ such that $|F(\mathbf{V}) - f(\mathbf{V})| < \epsilon$ for all $\mathbf{V} \in \Omega$.*

*Proof.* The idea is to write $F$ as a composition $F = \tilde{F} \circ \omega$ and $\omega = h \circ y$. We show that multiplication with $\mathbf{W}_h$ and and taking the L$_2$-norm can compute $y$, and that the dense network with one hidden layer can approximate $\tilde{F} \circ h$.

Call an element $\mathbf{V} \in \Omega^\nu$ *oriented* if $\boldsymbol{v}_1 = x_1 \mathbf{e}_x$, $\boldsymbol{v}_2 = x_2 \mathbf{e}_x + y_2 \mathbf{e}_y$, and $\boldsymbol{v}_3 = x_3 \mathbf{e}_x + y_3 \mathbf{e}_y + z_3 \mathbf{e}_z$, with $x_1, y_2, z_3 > 0$. Define $\omega : \Omega^\nu \to \mathbb{R}^{3\nu - 3}$ to be the *orientation function* that orients its input and then extracts the vector of $3\nu - 3$ coefficients, $[x_1, x_2, y_2, x_3, y_3, z_3, \ldots, x_i, y_i, z_i, \ldots]^T$. These

elements can be written as

$$
\begin{array}{rcl}
x_1 & = & \|\boldsymbol{v}_1\|_2 \\
x_i & = & \boldsymbol{v}_i \cdot \boldsymbol{v}_1 / x_1, \quad i \geq 2 \\
y_2 & = & \sqrt{\|\boldsymbol{v}_2\|_2^2 - x_2^2} \\
y_i & = & (\boldsymbol{v}_i \cdot \boldsymbol{v}_2 - x_i x_2)/y_2, \quad i \geq 3 \\
z_3 & = & \sqrt{\|\mathbf{v}_3\|_2^2 - x_3^2 - y_3^2} \\
z_i & = & (\boldsymbol{v}_i \cdot \boldsymbol{v}_3 - x_i x_3 - y_i y_3)/z_3, \quad i \geq 4
\end{array}
$$

and are invariant under rotation and reflection, because they are defined using only the norms and inner products of the $\mathbf{v}_i$. Then $F = \tilde{F} \circ \omega$, where $\tilde{F} : [-b, b]^{3\nu - 3} \to \mathbb{R}$.

The key insight is that if we construct $\mathbf{W}_h$ such that the rows of $\mathbf{W}_h \mathbf{V}$ are the original vectors $\mathbf{v}_i, \forall i$ and all differences $\mathbf{v}_i - \mathbf{v}_j, \forall i, j \leq \min(i, 3)$, then we can compute $\omega(\mathbf{V})$ from the row-wise norms of $\mathbf{W}_h \mathbf{V}$. That is, $\omega = h \circ y$ where $\mathbf{y} = y(\mathbf{V}) = \|\cdot\|_2 \odot (\mathbf{W}_h \mathbf{V}) \in \mathbb{R}^{4\nu - 6}$ and $h$ is an application of the cosine law. The GVP precisely computes $\mathbf{y}$ as an intermediate step: we can write $G_s(\mathbf{V}) = \sigma \odot (\mathbf{W}_m \mathbf{y} + \mathbf{b})$. It remains to show that there exists a form $\tilde{f}(\mathbf{y}) = \mathbf{w}^T [\sigma \odot (\mathbf{W}_m \mathbf{y} + \mathbf{b})]$ that $\epsilon$-approximates $\tilde{F} \circ h : [-2b, 2b]^{4\nu - 6} \to \mathbb{R}$. Up to a translation and uniform scaling of the hypercube, this is the result of the Universal Approximation Theorem (Cybenko, 1989). $\qquad \square$

## B  SYNTHETIC TASKS

We perform controlled experiments on a synthetic dataset in order to analyze the benefits of the GVP architecture and determine if it indeed improves the geometric and joint geometric-relational reasoning abilities of GNNs.

**Dataset**  The synthetic dataset is designed to mimic the essential qualities of the domain of protein structures. Each "structure" consists of $n = 100$ random points in $\mathbb{R}^3$, distributed uniformly in the ball of radius $r = 10$, with the constraint that no two points are less than distance $d = 2$ apart. Each position is also associated with a random unit vector (a "sidechain") to endow it with an orientation. Three points are randomly chosen and are labelled as "special"; these will be used to define the learning tasks. We generate 20k "structures" and split them 80% train : 10% validation : 10% test.

In the voxelized representation of a data point, the volume is voxelized into unit cubes. Each point is partitioned into a neighborhood of eight voxels by trilinear interpolation, such that exact coordinate information is retained. Separate channels are used for the special and non-special points. The "sidechains" are represented with a set of $n = 100$ points located at the ends of the unit vectors and mapped into a third channel.

In the graph-structured representation of a data point, a proximity graph is drawn with $k = 10$ nearest neighbors. Each node is labelled with a one-hot encoding of its type ("special" or "non-special") and each edge with its Euclidean length. In the vanilla GNN, orientation information is encoded by additionally including all three dot products in each edge embedding. In the GVP-GNN, each node embedding contains the node's "sidechain" vector, and each edge embedding contains a unit vector indicating the direction of the edge.

**Tasks**  We identify two regression tasks to exemplify geometric and relational reasoning, respectively. In the "Off-center" task, the network predicts the distance from the centroid of the three special points to the centroid of the entire structure. In the "Perimeter" task, the network predicts the perimeter of the triangle defined by the three special points. We characterize the former as primarily geometric, as it requires reasoning about global properties of the 3D *shape*, in particular points in space that are not themselves nodes, and the latter as primarily relational, as it involves distances between three specific *pairs* of nodes. Finally, to represent a problem with geometric *and* relational aspects, in the "Combined" task we attempt to predict the difference of the (normalized) off-center and perimeter objectives.

**Models** We train a 3-layer shallow CNN and a 3-layer GNN with a single-layer feed-forward network. These are compared against a GVP-GNN that is otherwise identical to the standard GNN. To reflect the spirit of the synthetic experiment, all models have the same intermediate dimensionality of 32 (4 vector and 20 scalar channels in the GVP-GNN), we use the same training procedure for all models, and no hyperparameter tuning or architecture search is performed.

**Results** The results of the synthetic experiments are shown in Table 5. The vanilla GNN significantly outperforms the CNN on the perimeter task, while the CNN significantly outperforms the GNN on the off-center task, supporting our conceptual framework of the relative strengths of the two architectures. However, the GVP-GNN matches (and even outperforms) the CNN on the geometric task while maintaining the GNN's performance on the relational task. It additionally significantly outperforms *both* models on the combined task. On the basis of these results, the GVP appears successful in combining the strengths of the CNN and GNN into a single architecture.

Table 5: Performance of the three compared model architectures on the off-center (geometric), perimeter (relational), and combined objectives. The MSE losses are standardized such that predicting a constant value (i.e. the mean) would result in unit loss. Results are reported as the mean $\pm$ S.D. over $k = 5$ random splits, where the best of three random seeds is taken for each split.

| Model | Parameters | Off-center (geometric) | Perimeter (relational) | Combined |
|---|---|---|---|---|
| CNN | 59k | $0.319 \pm 0.014$ | $0.532 \pm 0.028$ | $0.522 \pm 0.016$ |
| GNN | 40k | $0.871 \pm 0.045$ | $0.128 \pm 0.009$ | $0.421 \pm 0.025$ |
| GVP-GNN | 22k | $\mathbf{0.206 \pm 0.024}$ | $\mathbf{0.106 \pm 0.006}$ | $\mathbf{0.155 \pm 0.024}$ |

## C  MQA DATASETS: FURTHER DETAILS

The MQA training and validation dataset includes 528 targets from CASP 5-10 and 150 candidate structures per target. These targets are partitioned at random into 480 training targets and 48 validation targets. We include native structures for training and validation to make use of the greatest range of GDT-TS scores. We do not include native structures for testing in order to mimic CASP and real-world applications and because other methods were not tested on native structures.

In the CASP assessments, stage 1 refers to a set of 20 candidate structures per target and stage 2 to a set of 150 candidate structures per target (5 from each structure prediction server). Both sets are pre-designated by the CASP organizers.

There has been slight inconsistency in the literature with regards to the exact composition of the CASP 11 and 12 test sets. We use the list established by Karasikov et al. (2019) because nearly all recent methods have been benchmarked on this set at some point. The CASP 13 test set includes 1472 stage 2 candidate structures from the following 20 targets: T0950, T0951, T0953s1, T0953s2, T0954, T0955, T0957s1, T0957s2, T0958, T0960, T0963, T0966, T0968s1, T0968s2, T1003, T1005, T1008, T1009, T1011, T1016. These were the targets for which candidate structures, submitted predictions, and ground-truth scores were publicly available (obtained as described by Baldassarre et al. (2020)) at the time of writing. The exact numbers of targets and structures in each set can be found in Table 6.

Table 6: MQA datasets

| Dataset | # Targets | # Structures | Includes natives? |
|---|---|---|---|
| Training | 480 | 72000 | Yes |
| Validation | 48 | 7200 | Yes |
| CASP 11 stage 1 | 84 | 1680 | No |
| CASP 11 stage 2 | 83 | 12450 | No |
| CASP 12 stage 1 | 40 | 800 | No |
| CASP 12 stage 2 | 40 | 5950 | No |
| CASP 13 stage 2 | 20 | 1472 | No |

## D    TRAINING AND HYPERPARAMETERS

To train the MQA model to perform regression against the model quality score, we use a sum of an absolute loss and a pairwise loss. That is, for each training step we intake pairs $i, j$ where $i, j$ are candidate structures for the same target and compute

$$\mathscr{L} = H(y^{(i)} - \hat{y}^{(i)}) + H(y^{(j)} - \hat{y}^{(j)}) + H\left((y^{(i)} - y^{(j)}) - (\hat{y}^{(i)} - \hat{y}^{(j)})\right) \tag{6}$$

where $H$ is the Huber loss. When reshuffling at the beginning of each epoch, we also randomly pair up the candidate structures for each target. Interestingly, adding the pairwise term also improves global correlation, likely because the much larger number of possible pairs makes it more difficult to overfit.

To train the CPD model to perform classification / discrete generative modelling, we use the cross-entropy / negative log likelihood loss.

For both the MQA and CPD model, we use node and hidden embeddings with 16 vector and 100 scalar channels and edge embeddings with 1 vector and 32 scalar channels. The input node and edge features are first transformed by a sequence of GVPs to these dimensionalities before graph propagation. In all training runs, we use the Adam optimizer to perform mini-batch gradient descent. Batches are constructed by grouping structures of similar size to have a maximum of 1800 residues per batch for CPD and 3000 residues per batch for MQA. We also tune the following hyperparameters over a total of 70 training runs:

- Learning rate in the range of $10^{-4}$ to $10^{-3}$
- Dropout probability in the range of $10^{-4}$ to $10^{-1}$
- Number of graph propagation layers in the range of 3 to 6
- Relative weight of the MQA pairwise loss in the range of 0 to 2

All models are implemented in TensorFlow 2.1 and trained for a maximum of 100 epochs. This takes around two days for both models on a single Titan X GPU. However, we note that the GPU memory, not compute power, is the bottleneck when training, based on the the average volatile GPU usage. We therefore anticipate that the runtime can be further optimized.

## E    ADDITIONAL RESULTS

### E.1    DIMENET ON MQA

DimeNet (Klicpera et al., 2019) is a recent GNN architecture designed to incorporate the 3D geometry of small molecule graphs by encoding relative edge orientations in a local spherical Bessel basis. DimeNet and our architecture are similar in that both seek to leverage geometric aspects of a problem domain on top of graph-structured representations. However, unlike our architecture, Dimenet uses rotation-invariant features to indirectly encode geometry into its message-passing operations. Additionally, it updates *edge* embeddings by propagating messages between each pair of neighboring edges. While this paradigm appears well-suited for the domain of learning from small molecules, it does not scale well to large protein structure graphs. In evaluating DimeNet on MQA, we could only extend the distance cutoff to 7.5 angstroms, while 30 neighbors corresponds to roughly 13 angstroms. DimeNet does not perform comparably to our model, or to previous GNNs designed for learning from structure such as GraphQA (Table 7).

### E.2    MQA: RESULTS ON CASP 13

We report results for all 23 single-structure methods assessed in CASP 13 for which scores on all 20 targets are available (Table 8). The following 7 methods were excluded because they do not report results for some targets: LamoureuxLab, SBROD-server, SBROD, 3DCNN, MESHI-server, SBROD-plus, FALCON-QA, and Grudinin. We do include comparisons with LamoureuxLab (previously 3DCNN), 3DCNN (previously Ornate), and SBROD on CASP 11-12. All of the methods highlighted as top-performing by the CASP organizers in Cheng et al. (2019) are in our comparison for CASP 13. All predictions were obtained from the CASP download center as described by Baldassarre et al. (2020).

Table 7: Comparison of our GVP architecture, DimeNet, and the GraphQA, another GNN-based MQA method, on CASP 11-12. As in the main text, the global and mean per-target Pearson correlations are shown. DimeNet does not perform comparably to either GVP-GNN or GraphQA.

| Method | CASP 11 Stage 2 | | CASP 12 Stage 2 | |
|---|---|---|---|---|
| | Global | Per-target | Global | Per-target |
| GVP-GNN | **0.87** | **0.45** | **0.82** | **0.62** |
| GraphQA | 0.82 | 0.38 | 0.81 | 0.61 |
| DimeNet | 0.61 | 0.30 | 0.62 | 0.47 |

Table 8: Comparison of GVP-GNN against all 23 available single-structure MQA methods in CASP 13 sorted by global correlation. The total number of predictions is shown, which may be less than 1472 even though they include all 20 targets. GVP-GNN is the best-performing method in terms of global correlation. In terms of per-target correlation, GVP-GNN outperforms all other structure-only methods and also all methods using sequence profiles except for ProQ4 and two ProQ3D variants.

| Method | Global | Per-target | Predictions | Structure only? |
|---|---|---|---|---|
| GVP-GNN | 0.888 | 0.671 | 1472 | Yes |
| ProQ3D | 0.847 | 0.660 | 1467 | No |
| SASHAN | 0.840 | 0.633 | 1472 | Yes |
| MESHI-corr-server | 0.838 | 0.651 | 1472 | Yes |
| ProQ3 | 0.822 | 0.576 | 1468 | No |
| MESHI | 0.813 | 0.666 | 1472 | Yes |
| MESHI-enrich-server | 0.813 | 0.666 | 1472 | Yes |
| FaeNNz | 0.810 | 0.650 | 1472 | Yes |
| ProQ3D-CAD | 0.803 | 0.673 | 1468 | No |
| ProQ3D-lDDT | 0.803 | 0.687 | 1467 | No |
| ProQ2 | 0.802 | 0.577 | 1472 | No |
| ProQ3D-TM | 0.791 | 0.654 | 1467 | No |
| MASS1 | 0.776 | 0.582 | 1472 | Yes |
| VoroMQA-A | 0.744 | 0.595 | 1472 | Yes |
| VoroMQA-B | 0.726 | 0.586 | 1472 | Yes |
| MASS2 | 0.689 | 0.584 | 1472 | Yes |
| MULTICOM-NOVEL | 0.652 | 0.551 | 1472 | No |
| ProQ4 | 0.604 | 0.691 | 1472 | No |
| PLU-AngularQA | 0.577 | 0.460 | 1472 | Yes |
| Bhattacharya-Server | 0.577 | 0.501 | 1452 | No |
| Bhattacharya-SingQ | 0.498 | 0.525 | 1452 | No |
| Kiharalab | 0.375 | 0.565 | 1472 | No |
| PLU-TopQA | 0.239 | 0.049 | 1472 | Yes |
| Jagodzinski-Cao-QA | 0.180 | 0.341 | 1472 | Yes |

### E.3 CPD: Results on TS50

We compare against a number of recent CPD methods on the TS50 test set in Table 9.[10] These include two CNNs (ProDCoNN and DenseCPD), a distance-map method (SBROF), and sequential representation methods (Wang's model and SPIN2). We also evaluate the GNN ProteinSolver on TS50 by sampling 100 sequences with temperature 1 (the default setting) for each structure using the public web server. No canonical training and validation sets exist for TS50. Therefore, in order to evaluate on TS50, we remove sequences with more than 30% similarity from the CATH 4.2 training and validation sets and retrain our model. We outperform all other methods with the exception of DenseCPD, a CNN method with canonical orientations. Interestingly, DenseCPD leverages the same underlying representation as ProDCoNN, yet achieves remarkably better performance. The main

---

[10]Except for ProteinSolver, data for other methods is obtained from Qi & Zhang (2020) and Li et al. (2014).

difference between the two methods is that ProDCoNN has 4 convolutional layers and DenseCPD has 21 layers organized into dense residual blocks.

Table 9: Sequence recovery on TS50. Recovery for GVP-GNN and ProteinSolver is as defined in Table 1; recovery for other methods, which model residues independently, is just classification accuracy. GVP-GNN is the second best-performing method, behind the CNN method DenseCPD. There is no canonical training and validation set for methods evaluated on TS50.

| Method | Recovery % |
|---|---|
| GVP-GNN | 44.9 |
| DenseCPD (Qi & Zhang, 2020) | **50.7** |
| ProDCoNN (Zhang et al., 2019) | 40.7 |
| SBROF (Chen et al., 2019) | 39.2 |
| SPIN2 (O'Connell et al., 2018) | 33.6 |
| Wang's model (Wang et al., 2018) | 33.0 |
| ProteinSolver (Strokach et al., 2020) | 30.8 |
| SPIN (Li et al., 2014) | 30.3 |
| Rosetta | 30.0 |

## F  VISUALIZATION AND INTERPRETATION OF LEARNED FEATURES

The geometric vector perceptron updates the vector features, in addition to scalar features, at node embeddings during graph propagation. Therefore, while the *input* vector channels represent the forward and reverse directions at each amino acid, the *intermediate* layers represent learned vector features. Could some of these features correspond to interpretable properties of the structure? Among a total of 64 intermediate vector channels learned by the MQA model, a few appeared visually interpretable and are shown on selected structures in Figure 2. We caution against generalizing from the necessarily small number of images that could be manually inspected, but find these preliminary visualizations intriguing.

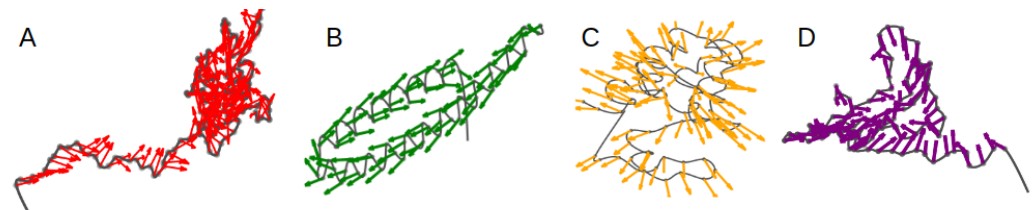

Figure 2: Four different learned vector channels of the MQA model are visualized on four separate structures. The backbone is represented as a chain of points, and each vector is rooted at the position of the amino acid node to which it belongs. From left to right: the vectors appear to A) point in the direction of motion that would make the protein more compact; B) point along the central axis of the alpha helix; C) point outwards from the structure; and D) point inwards into the structure.

