# OpenReview forum: "Learning from Protein Structure with Geometric Vector Perceptrons"
_ICLR.cc/2021/Conference — ICLR 2021 Spotlight_

### Official Review · AnonReviewer1 · 2020-10-25
**Interesting paper introducing GVP architecture and showing that it results in good performance for relevant tasks**

**Rating:** 7
**Confidence:** 3

**Review:**

The challenge of predicting the structure of biological macro-molecules is widely relevant in many applications and difficult to address. This paper divides the types of approaches taken to address this challenge into those that use "geometric" information (i.e. positions of molecules in space), and those that utilize "relational methods" mainly through graphs (how different parts of molecule relate). This study is an attempt to integrate the two source of information by a novel network architecture. They introduce geometric vector perceptrons as a way of summarizing geometric information for graph layers without loss of information as it happens in dense layers. They evaluate the performance of these architectures on MQA and CPD tasks, both relevant and standard benchmarks in the field.

Positives:

- The paper well-written and sets up the problem and how it supposed to be solved very well.

- The method seems to be high-performing on well-established benchmarks.

- The conceptual contribution of GVP seems promising, intuitively relevant and usable.

- Ablation and hyper-parameter studies are well-scoped.

Areas for improvement:

- The authors advertise GVPs as a "drop-in replacement" for MLP layers, however, looking at their ablation studies, while the MLP layer is less performant than GVP, it still is outperforming many SOTA methods. For this reason I suggest including the MLP results in the main text, so that the performance improvement is clear. It is also relevant to discuss what drives the performance of the method such  that even with MLP, it outperforms sophisticated architectures introduced before.

- For MQA tasks, A discussion and ideally quantitative comparison with Ingraham 2019 ICLR (Differentiable Simulator), and AlphaFold (Senior et al 2020 Nature) seems pertinent. At least to me it is not immediately obvious if learning seq -> structure is a sufficiently different task not to be considered/discussed here. Presumably sequence data is more widely available than the geometry.

---

> ### Author Response · Authors · 2020-11-24
> **Author Response to Reviewer 1**
>
> We thank the reviewer for the positive evaluation of our work and recognition of our strong empirical results, intuitive appeal of our method, and ablation studies, and appreciate the suggestions for further improvements.
>
> **Ablation studies**: We follow the reviewer’s suggestion and now present the full results of the ablation study in the main text (Table 4, Section 5) along with a unified discussion of these results. For both CPD and MQA, we note that replacing the GVP layers with MLP layers leads to a tangible decrease in performance and the resulting ablated models do not outperform state-of-the-art methods. Specifically, they do not outperform GraphQA and Structured GNN, the previous best methods on MQA and CPD. It is true that these methods, both of which are GNNs, do perform better than some other methods shown in the tables, and we also find this intriguing. Our main claim is that without the GVP, our method performs no better than previous baseline GNNs.
>
> **MQA comparisons**: We thank the reviewer for bringing up the relation between model quality assessment and structure generation. Both are important problems in the protein structure prediction process, but their formulation and role are different. Structure generation methods (such as Ingraham et al. 2019 ICLR and Senior et al. 2020 Nature) map sequences to a generative model over 3D structures, while MQA methods map 3D structures (produced by structure generation methods) to quality scores. MQA is a crucial step in the structure prediction pipeline, as picking out highly accurate models from a large pool of candidate structures is often a performance bottleneck. CASP, the community experiment which biennially evaluates methods for structure prediction, recognizes this by establishing MQA as its own track, with its own characteristic performance metrics and problem definitions. Therefore, a direct comparison with methods that model 3D structure from sequence, such as Ingraham et al. 2019 ICLR and Senior et al. 2020 Nature, is difficult to formulate. The revised manuscript does contain comprehensive comparisons with MQA methods from the most recent CASP experiment for which results are available (CASP 13).

---

### Official Review · AnonReviewer4 · 2020-10-28
**The paper addresses an important issue in computational biology. The paper is well written and steps are well explained.**

**Rating:** 10
**Confidence:** 4

**Review:**

In this paper, the authors introduce a novel procedure to predict or acquire insights from the structure of a macromolecule (such as a protein, RNA, or DNA), represented as a set of positions associated with atoms or groups of atoms in 3D Euclidean space. Their approach, called GVP-GNN, can be applied to any problem where the input domain is a structure of a single macromolecule or molecules bound to one another. Their approach is divided into two steps: model quality assessment and computational protein design.
The paper addresses a notable problem in computational biology: learning on 3D structures of large biomolecules.  I have noticed various industry-based studies struggling to find the right solution.
The paper is powerfully written, and the approach is novel. The steps are described explicitly, which helps the reader to understand them.

I have a few suggestions for the Authors. Please grammar proof the paper thoroughly to avoid small grammatical mistakes such as articles, or typos.
In the results, please refer to the suggested approach as GVP-GNN. Sometimes, you refer to your method as "ours". Please be uniform.
The availability of a dataset, especially the synthetic dataset, will allow the paper to be reproducible by others.

---

> ### Author Response · Authors · 2020-11-24
> **Author Response to Reviewer 4**
>
> We thank the reviewer for the recognition of the strength of our architecture, in particular its novelty, intuitive appeal, and broad applicability to an important and difficult class of scientific problems. We agree that leveraging 3D biomolecular structure in machine learning remains a large challenge and appreciate the recognition of our contributions.
>
> We have uploaded our synthetic dataset to Google Drive (anonymous account) and included a link in Appendix B. At the time of the camera-ready version, we will also release our complete training, validation, and testing datasets for MQA and CPD, as well as our codebase for building, training, and testing the model.
>
> We have aimed to remove any remaining grammatical and typographical errors and have updated all sections to consistently refer to our method as GVP-GNN.

---

### Official Review · AnonReviewer2 · 2020-10-28
**Applying a GNN with a modified layer on protein backbones for quality assessment and design**

**Rating:** 6
**Confidence:** 4

**Review:**

This work uses a graph representation of the protein backbone and a GNN
for model quality assessment (MQA) and protein design.

Strengths:
The GNN architecture proposed has the important property that the vector and scalar outputs
are equivariant and invariant with respect to composition of 3D rotations and reflections.

The submission is clear and correct, and the experiments are reproducible.

Weaknesses:
Most datasets used in this work and methods compared with are from 2016 or earlier (CASP12 or earlier, and for example ProQ3D);
whereas a more relevant comparison would use datasets from CASP13 (or even 14) and recent methods (for example ProQ4).
Nearly none of the methods in CASP 12 (2016) and earlier CASPs used deep learning
since Tensorflow was released in 11.2015 and PyTorch in 9.2016.
Only in CASP13 (2018) did the majority of groups use deep learning.
There is an inconsistency between the text describing the dataset used for MQA in section 4 and Table 6 in Appendix C, which may be clarified in the text by mentioning CASP13.

While the work does compare against several GNNs, a comparison with important recent work on protein design using GNNs
such as ProteinSolver [1] or a reference would improve the paper.
Other missing references on protein design are methods using 3D CNNs [2] conditional VAE [3] and conditional GAN [4],
as well as work on MQA [5]. Therefore, the unequivocal conclusion that the method "empirically outperforms existing architectures on learning quality scores and sequence designs, respectively, from protein structure" is partially accurate and may be rephrased to improve the paper.

[1] Fast and Flexible Protein Design Using Deep Graph Neural Networks, Strokach et al, 2020.
http://design.k8s.proteinsolver.org
[2] A structure-based deep learning framework for protein engineering, Shroff et al, 2019.
[3] Design of metalloproteins and novel protein folds using variational autoencoders, Greener et al, 2018.
[4] Structural bioinformatics de novo protein design for novel folds using guided conditional Wasserstein generative adversarial networks, Karimi et al, 2019.
[5] Deep transfer learning in the assessment of the quality of protein models, Hurtado et al, 2018.
https://github.com/ElofssonLab/ProQ4

---

> ### Author Response · Authors · 2020-11-24
> **Author Response to Reviewer 2**
>
> We thank the reviewer for recognizing the strengths of our architecture and the reproducibility of the empirical results. As part of the revisions, we have added additional experiments for both MQA and protein design as well as references to relevant recent work to the manuscript.
>
> **MQA results**: We enthusiastically agree that further comparisons on CASP 13 would strengthen the manuscript. We have included an extensive list of additional comparisons for CASP 13 as part of the revisions (Section 5, Table 2 and Appendix E, Table 8). We report results for a total of 23 methods. The comparison includes all single-structure methods highlighted in the CASP 13 special issue in PROTEINS [1], among them ProQ4. Our method is the top-performing single-structure method on CASP13 in terms of global correlation and also outperforms all structure-only methods in terms of per-target correlation.
>
> **Recency of MQA methods**: We agree that it is important to compare with recent methods. All of the MQA methods we originally compare against were published in 2017 or later, significantly more recent than CASP 12. In particular, SBROD and GraphQA first appeared in 2019. We use the CASP 11/12 dataset mainly because these methods were originally benchmarked on that dataset. Our new CASP 13 results also include many other recent methods, including some published in 2020.
>
> **Deep learning MQA methods**: We agree it is important to compare against other deep learning methods. We have added comparisons to many deep learning methods in our CASP 13 results (FaeNNz, Multicom-Novel, ProQ4, AngularQA, TopQA, Bhattacharya). Additionally, all of the methods in our original comparison, with the exception of SBROD and VoroMQA, use deep networks (dense, convolutional, or graph neural networks).
>
> We have added a discussion of the CASP 13 dataset in Section 4 to present all rows from Table 6 (Appendix C) in the main text.
>
> **Protein design**: We thank the reviewer for bringing the mentioned relevant works to our attention. We have added references and a brief discussion of ProteinSolver in Section 2. Additionally, using the provided web server link, we have included a comparison with ProteinSolver on the TS 50 test set in Appendix E, Table 9. We report better results for our method GVP-GNN. We reference and briefly discuss the other mentioned works in Section 2 of the updated manuscript. We note that our discussion of related work remains organized by architecture type to best align with the main focus of our technical contribution.
>
> [1] Cheng, et al. Estimation of Model Accuracy in CASP13. Proteins: Structure, Function, and Bioinformatics, 87(12):1361–1377, 2019.

---

### Official Review · AnonReviewer3 · 2020-10-29
**The idea that leverages the geometric and relational aspects of the 3D structure is novel. However, some details of the approach are not clear, which hurt the readability of the paper.**

**Rating:** 6
**Confidence:** 4

**Review:**

This paper aims to leverage the geometric and relational aspects of the 3D structure
simultaneously with the proposed GVP layer. The approach extends standard dense layers. The
authors claim that GNNs with such layers can perform both geometric and relational reasoning
on representations of macromolecules. The experiments on two problems in protein structure
learning show improvement over CNN and GNN. However, some details are missing, and the
intuition of the methods is not clear. The draft needs to be refined and organized well.
Pros:
1. The proposed GVP can learn geometric features from protein structure and is
rotation/reflection-invariant.
2. The learned vector features in the intermediate layer are interpretable.

Cons:
1. Fig.1 A is not clear. The intermediate layer and weight matrices are all represented with
blocks. The schematic diagram needs to be refined.
2. Some equations are not clear. In equation (2) and (3), the subscripts j, m, e, v are not
defined.
3. The writing and format of the paper are hard to read. In the 3.2 section, lots of symbols
are used without definition (e.g. C, $alpha$, RBF). In the 3.3 section, h_v^j, h_v^i is not
clear. some schematic diagrams will be helpful for illustrating the network structure.
4. The intuition and motivation of the methods are not clear. For example, why is it easier
for GNN to access the global geometric properties of protein structure with GVP?

---

> ### Author Response · Authors · 2020-11-24
> **Author Response to Reviewer 3**
>
> We thank the reviewer for the positive evaluation, recognition of our empirical results, and suggestions to further improve our manuscript. We have revised the manuscript with the goal of making the representation of our method clearer and more accessible.
>
> **Figure 1a**: We have refined the schematic illustration of our method in Figure 1a. The updated figure now shows operations and features using distinct graphical representations and we also depict vector-valued and scalar-valued features differently. We further highlight what parts of the geometric vector perceptron correspond to its scalar and vector channel, respectively.
>
> **Equations**: We have added information to section 3.2 and section 3.3 to clarify the meaning of symbols and acronyms. We have also made changes to the way Equations 2-4 are written to improve clarity and added further text explaining the represented quantities and notations used.
>
> **Network structure**: We have added text in section 3.3 to clarify how geometric vector perceptrons are used to update node features as part of the message-passing algorithm in the graph neural network.
>
> **Intuition**: We have refined the beginning of Section 3 and the description of the synthetic tasks (Appendix B) to provide further motivation for our method and examples for why the GVP helps the GNN access geometric information. At a high level, a standard GNN with scalar-invariant representations of geometry must transform between local coordinates to propagate vector-valued geometric information across the graph---for example, the representation of an arbitrary position in space. With the introduction of geometric vector features, this information can be represented directly as a vector and propagated using simple vector addition.

---

### Author Response · Authors · 2020-11-24
**Summary**

We sincerely thank all reviewers for their thorough evaluation and suggestions for further improvement. We in particular appreciate that the reviewers recognized the **novelty** (R1, R3, R4) of our method and how its **important equivariant properties** (R2, R3) and **relevant conceptual contribution** (R1) help to address a **notable problem** (R4) in computational biology. The reviewers also point out that our method is **high-performing** on **well-established benchmarks** (R1), considers **important and difficult tasks** (R1, R4), and is furthermore **broadly applicable** (R4). We are glad that the reviewers consider our submission **clear and correct** (R2), the **experiments reproducible** (R2), and the **ablation studies well-scoped** (R1). R1 and R4 consider the manuscript **well written** and R3 highlights the **interpretability of the learned features.**

We have revised our manuscript based on the insightful feedback provided. We address the reviewers’ comments and concerns in individual responses to each reviewer below.

At a high level, in our revisions, we have focused on four aspects:
1) Improving the clarity of the method’s presentation (Section 3), including a revised schematic figure (Figure 1a)

2) Performing additional experiments and comparisons for model quality assessment, specifically for CASP 13 (Table 2, Section 5 and Table 8, Appendix E). The new results continue to support the strengths of our architecture.

3) Providing further intuition/motivation for the architecture (beginning of Section 3)

4) Discussing the full ablation results in the main manuscript (Table 4, Section 5)

---

### Comment · ~Lei_Wang21 · 2021-08-26
**GVP seems not to prove transformation equivariance**

Here, the paper proposed the a novel method of macromolecules (proteins) named GVP and claimed the vector and scalar outputs of the GVP are equivariant and invariant, respectively, with respect to an arbitrary composition of rotations and reflections in 3D Euclidean space. In fact,  the transformation of macromolecules (proteins) includes rotation, translation, reflections. In the GVP, translation can not be guaranteed to be equivariant and invariant. So, i think the paper did not handle the translation from Protein Structure.

---

> ### Comment · ~B_Jing1 · 2021-09-13
> **Translation equivariance**
>
> Thank you for your comment. The translation equivariance of our method is manifested at the GVP-GNN level rather than the GVP level. All vector features passed into the GVP as part of graph propagation are derived from relative position vectors, and these relative position vectors are automatically invariant to translations of the protein structure. This is the same sense of translation-equivariance common to many point-cloud based ML methods; for example, [Tensor Field Networks](https://arxiv.org/abs/1802.08219).

---

### Decision · Program_Chairs · 2021-01-07
**Final Decision**

**Decision:**

Accept (Spotlight)

**Comment:**

This work uses a graph representation of the protein backbone and a GNN for model quality assessment (MQA) and protein design. The proposed GNN has the property that the vector and scalar outputs are equivariant and invariant with respect to composition of 3D rotations and reflections. Overall speaking, the reviewers like this paper very much (especially its technical novelty), and provide quite positive comments. On the other hand, there are also some concerns being mentioned:

1)	The datasets used in the experiments are a little old – experiments on CASP 13 are preferred.
2)	Some technical details are not very clear and the paper writing needs improvements
3)	Experimental comparison with some recent baselines is missing.

The authors did a good job in their rebuttal and paper revision. Most of the above concerns have been addressed. Therefore, we think the current version of the paper is clearly beyond the bar of ICLR.